# Differential immunomodulation of porcine bone marrow derived dendritic cells by *E. coli* Nissle 1917 and β-glucans

Mirelle Geervliet[1], Laura C. P. Lute[1], Christine A. Jansen[2], Victor P. M. G. Rutten[2,3], Huub F. J. Savelkoul[1], Edwin Tijhaar[1]*

**1** Cell Biology and Immunology group, Department of Animal Sciences, Wageningen University, Wageningen, The Netherlands, **2** Department of Infectious Diseases and Immunology, Faculty of Veterinary Medicine, Utrecht University, Utrecht, the Netherlands, **3** Department of Veterinary Tropical Diseases, Faculty of Veterinary Science, University of Pretoria, Pretoria, South Africa

* edwin.tijhaar@wur.nl

**Data Availability Statement:** All data and files related to the manuscript will be available from 4TUCentre for Research (Wageningen University).

## Abstract

In early life and around weaning, pigs are at risk of developing infectious diseases which compromise animal welfare and have major economic consequences for the pig industry. A promising strategy to enhance resistance against infectious diseases is immunomodulation by feed additives. To assess the immune stimulating potential of feed additives *in vitro*, bone marrow-derived dendritic cells were used. These cells play a central role in the innate and adaptive immune system and are the first cells encountered by antigens that pass the epithelial barrier. Two different feed additives were tested on dendritic cells cultured from fresh and cryopreserved bone marrow cells; a widely used commercial feed additive based on yeast-derived β-glucans and the gram-negative probiotic strain *E. coli* Nissle 1917. *E. coli* Nissle 1917, but not β-glucans, induced a dose-dependent upregulation of the cell maturation marker CD80/86, whereas both feed additives induced a dose-dependent production of pro- and anti-inflammatory cytokines, including TNFα, IL-1β, IL-6 and IL-10. Furthermore, *E. coli* Nissle 1917 consistently induced higher levels of cytokine production than β-glucans. These immunomodulatory responses could be assessed by fresh as well as cryopreserved *in vitro* cultured porcine bone marrow-derived dendritic cells. Taken together, these results demonstrate that both β-glucans and *E. coli* Nissle 1917 are able to enhance dendritic cell maturation, but in a differential manner. A more mature dendritic cell phenotype could contribute to a more efficient response to infections. Moreover, both fresh and cryopreserved bone marrow-derived dendritic cells can be used as *in vitro* pre-screening tools which enable an evidence based prediction of the potential immune stimulating effects of different feed additives.

DOI: 10.4121/uuid:f27e43c4-ab38-41a2-be4d-c0753855b508.

**Funding:** This research was funded by The Dutch Research Council (NWO) and Vereniging Diervoeders Nederland (VDN). The award was received by H.F.J. (Huub) Savelkoul (grant number: 868.15.033, NWO Top Sector Project). The funders had no role in study design, data collection and analysis, decision to publish, or preparation of the manuscript. URL: https://www.nwo.nl/en.

**Competing interests:** The authors have declared that no competing interests exist.

## Introduction

Infectious diseases impact pig health and greatly impair animal welfare and efficiency of nutrient use, and thus animal performance [1]. To enhance resistance against infectious diseases, immunomodulation by feed additives may be a strategy to strengthen the pigs' immune competence. Feed additives that possess immune enhancing activity could prime cells of the immune system to respond more efficiently to infections. An important group of cells are professional antigen-presenting cells (APCs) like macrophages and dendritic cells (DCs). In particular, DCs are the key players in the initiation, differentiation and regulation of immune responses. In the gut, DCs sense and sample antigens from the gut luminal environment. Depending on the type of antigen encountered, DCs maturate and migrate towards the Mesenteric Lymph Node (MLN) where they interact with T- and B- cells [2, 3]. This makes DCs an important target for immunomodulatory approaches, including modulation by feed additives.

DCs identify conserved microbial molecules through pattern recognition receptors (PRRs), an interaction that may induce DC maturation and which is characterized by the upregulation of maturation markers (e.g. MHC-II, CD80 and CD86) [4]. In addition to these phenotypic changes, DC maturation results in the production of cytokines (e.g. TNFα, IL-1β, IL-6, IL-10 and IL-12) which play a key role in the development of both the innate and adaptive immune responses. As such, the DC maturation stage and the cytokine production profile direct immune responses. These markers and cytokines are therefore valuable parameters for the assessment of immunomodulatory effects of feed additives. Generation of porcine DCs to study these effects *in vitro* have only been reported in a small number of publications [5–7]. These DCs are generated from bone marrow (stem-) cells (BMDCs) or blood derived monocytes (MoDCs), and cultured in the presence of granulocyte macrophage colony-stimulating factor (GM-CSF) which is crucial for differentiation into immature DCs [8].

The first aim of this study was to assess immunomodulatory properties of two feed additives that differ in mode of action using *in vitro* cultured porcine BMDCs; yeast-derived β-glucans (MacroGard®) and the Gram-negative probiotic *E. coli* Nissle 1917 (EcN). MacroGard® is derived from the yeast cell wall of *Saccharomyces cerevisiae*, that contains a minimum of 60% β-1,3/1,6-glucans. β-Glucans are one of the most abundant forms of polysaccharides found inside the cell wall of bacteria, fungi (including yeast), and plants. They interact with receptors present on the cell surface of APCs, the primary receptor being Dectin-1 [9]. Especially β-glucans derived from fungi and yeast, which consist of a [1,3]-β-linked backbone with small numbers of [1,6]-β-linked side chains, are known for their immune stimulating effects [10–12]. Moreover, β-glucans exist in particulate and soluble forms, which can influence their immunomodulatory capacity. *Particulate β-glucans* induce stronger immune responses than *soluble β-*glucans by clustering of Dectin-1 receptors [13, 14]. The immune stimulating effects of particulate yeast-derived β-glucans and its use as a feed additive for pigs, makes MacroGard® a promising candidate for immunomodulation. Another promising candidate is EcN, a versatile Gram-negative probiotic strain with immunomodulating properties [15, 16]. Studies in humans and mice showed that EcN can interact with different receptors on APCs, the major receptor being Toll-like receptor 4 (TLR4). Crosslinking of these receptors on APCs enable the upregulation of cell maturation markers and production of cytokines [17]. Despite its use as a probiotic in humans for over a century, far less is currently known about the immune stimulating effects of EcN in pigs [18–20].

The second aim of this study was to compare porcine BMDCs derived from fresh (frhBMDCs) and cryopreserved (cryoBMDCs) bone marrow cells in their ability to assess the immunomodulatory properties of the selected feed additives as cryoBMDCs offer practical

advantages (e.g. storage). This study shows that β-glucans and EcN induce differential immunomodulatory effects in frhBMDCs and cryoBMDCs with regard to DC maturation and cytokine production. These results indicate that BMDCs can be used as a tool for large scale *in vitro* screening of feed additives with immunomodulatory potential.

## Materials and methods

### Ethics statement

This animal experiment was conducted in accordance with the Dutch animal experimental and ethical requirements and the project license application was approved by the Dutch Central Authority for Scientific Procedures on Animals (CCD) (Permit Number: AVD104002016515). The protocol of the experiment was approved by the Animal Care and Use committee of Wageningen University & Research (Wageningen, The Netherlands) (Protocol Number: 2016.W-0045.001). Pigs were euthanized with Euthasol® followed by exsanguination. Euthanasia was performed according to Good Veterinary Practice (GVP), and all efforts were made to minimize suffering.

### Generation and stimulation of BMDCs

Bone marrow was obtained from the tibia, femur and ribs of 4–6 weeks old Topigs Norsvin pigs as described previously [21]. Briefly, bones were flushed with RPMI 1640 Medium (with GlutaMAX™ supplement, Gibco™, MA, USA), containing 10% fetal calf serum (FCS, Gibco™), 1% L-Glutamine (Gibco™), 1% Penicillin Streptomycin (Pen/Strep, Gibco™) and 0.2% Normocin™ (InvivoGen, San Diego, CA), hereafter referred to as cell culture medium. Subsequently, bone marrow cells were either immediately used for frhBMDC culture or frozen overnight in FCS containing 10% Dimethylsulfoxide (DMSO, EMSURE®) at -80°C using Mr Frosty™ freezing containers (ThermoFisher Scientific™, Waltham, MA, USA). The following day, frozen bone marrow cells were stored in liquid nitrogen until further use. FrhBMDCs were cultured in a Corning® Costar® TC treated 96-well flat-bottom plate (Sigma-Aldrich, St. Louis, MO, USA) containing $1 \cdot 10^5$ cells/well in 200μl cell culture medium. For the cryoBMDCs culture, bone marrow cells were first thawed and quickly diluted (10 times) in cell culture medium and centrifuged at 200g for 6 minutes to remove DMSO. Subsequently, bone marrow cells were washed and cultured in a Corning® Costar® TC treated 96-well flat-bottom plate containing $1 \cdot 10^5$ cells/well in 200μl cell culture medium. For differentiation into immature BMDC (iBMDC), 20 ng/mL of recombinant porcine granulocyte macrophage colony-stimulating factor (rpGMCSF, R&D systems, Minneapolis, MN, USA) was added to the cell culture medium. In the first experiment (Figs 3 and 4) 25μl of cell culture medium (containing GM-CSF) was added on day 3 and on day 5, and differentiated iBMDC were stimulated on day 6 with various concentrations of LPS (10–0.001 μg/mL) for 24 hours. In the second experiment (Figs 5 and 6) 50μl of cell culture medium (containing GM-CSF) was added on day 3 and the iBMDC were stimulated with various concentrations of LPS (10–0.001 μg/mL), β-glucans (50 μg/mL– 1.563 μg/mL) or EcN (ratio 1:100–1:0.8) on day 4 for 24 hours. The morphology of the BMDCs was assessed every day by microscopy using the EVOS® FL Cell Imaging System (ThermoFisher Scientific™).

### Feed additives

For this study, the feed additives MacroGard® and *E. coli* Nissle 1917 were selected. MacroGard® (kindly provided by Orffa Additives B.V., Werkendam, The Netherlands) contains particulate β-1,3/1,6- glucans (minimum of 60% purity) derived from the yeast cell wall of

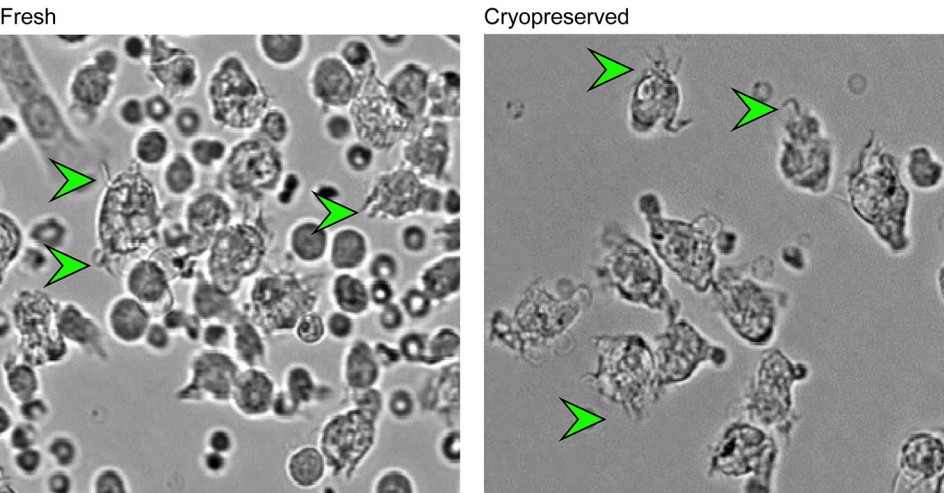

**Fig 1. Morphology of *in vitro* cultured frhBMDCs and cryoBMDCs.** Microscopic analysis of rpGM-CSF stimulated bone marrow cells differentiated into immature BMDCs at day 6 of the culture. Examples of BMDCs (approximately 15–30 μm in size) are indicated by arrowheads.

*Saccharomyces cerevisiae*. Other components include mannose (approximately 40% of the cell dry mass) and chitin (approximately 2% of the cell wall dry mass). The second feed additive is the Gram-negative probiotic strain *E. coli* Nissle 1917 (EcN, kindly provided by Ardeypharm, GmbH, Herdecke, Germany). In addition, lipopolysaccharide (LPS) from *Escherichia coli* (serotype O55:B5/L2880, Sigma-Aldrich) was selected as the positive control for this study. MacroGard® was diluted in milli-Q water containing 0.03M Sodium hydroxide (NaOH) and

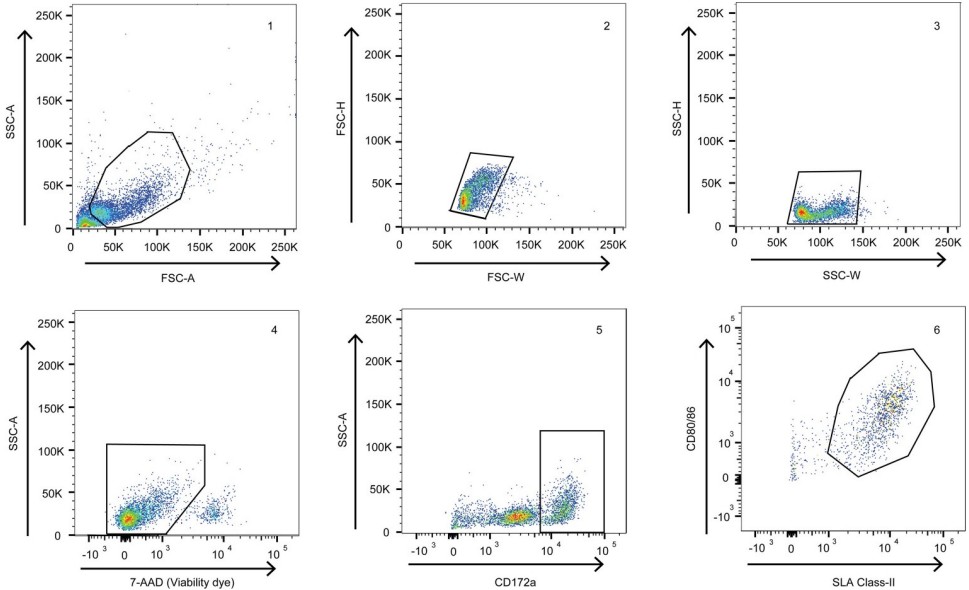

**Fig 2. Phenotype of freshly cultured porcine mononuclear phagocytes.** Gating strategy following multicolour flow cytometry staining using Abs against CD172a, SLA Class-II and CD80/86. Cells showing high forward scatter (FSC-A) and side scatter (SSC-A) profiles were gated, followed by the selection of single cells (FSC-W/H and SSC-W/H) and viable cells (SSC-A/7-AAD). Among these cells, BMDCs were defined as the CD172a$^{+/high}$ cells (SSC-A/CD172a) expressing SLA Class-II and CD80/86.

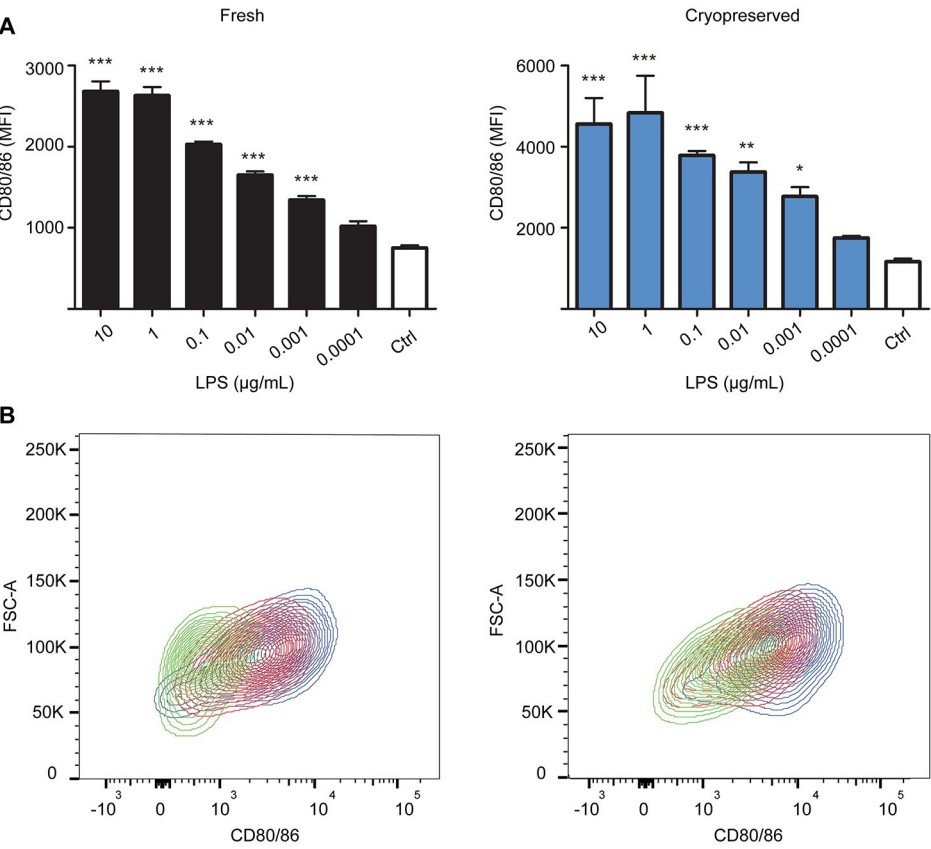

**Fig 3. FrhBMDCs and cryoBMDCs upregulate the maturation marker CD80/86 in a dose-dependent manner upon LPS stimulation.** (A) FrhBMDCs and cryoBMDCs (obtained from the same animal, n = 1) were stimulated with different concentrations of LPS or unstimulated using cell culture medium (negative control; Ctrl). After 24 hours, the expression (MFI) of the maturation marker CD80/86 was measured using Flow Cytometry. The data are shown as the means ± the standard error of the mean (SEM) of three technical replicates. A one-way ANOVA with a Dunnett's post hoc test was performed, comparing multiple groups to the untreated cells (control): *** = P<0.001, **P<0.01 and * P<0.05. (B) Representative contour plots of CD80/86 expression on LPS stimulated frhBMDCs and cryoBMDCs. The contour plots are based on forward scatter (y-axis) and CD80/86 expression (x-axis). Two different concentrations of LPS (10 μg/mL; blue and 0.01 μg/mL; red) and cell culture medium (negative control; green) are presented in this figure.

heated at 70˚C for 2.5 hours. Subsequently, the suspension was aliquoted and stored at -20˚C until use. To prevent large aggregates, the MacroGard® suspension was pushed ten times through a 0.8 x 50mm syringe-needle (Microlance™ 3) and ten times through a 0.5 x 16mm syringe-needle (Microlance™ 3) just before every experiment. To determine possible endotoxin contamination present in the β-glucan preparation an EndoZyme® test kit, which contains re-combinant factor C, was purchased from Hyglos (Hyglos GmbH, Bernried am Starnberger See, Germany) and performed according to the manufacturer's recommendations. In addition, HEK-293 cells expressing human TLR4 and harbouring a pNIFTY construct (Invivogen, Tou-louse, France) were used for the measurement of LPS induced NFκB activation after 24 hours. Both assays showed no evidence for endotoxin contamination (less than 0.06 endotoxin units or EU per mL) in 0.1g of the β-glucan preparation (S1 Fig).

The probiotic strain EcN was grown overnight under aeration to stationary phase at 37˚C in Lysogeny Broth (LB) medium. The bacteria were recovered by centrifugation (4000 g for 1 minute) and washed three times in RPMI 1640 Medium (with GlutaMAX™ supplement,

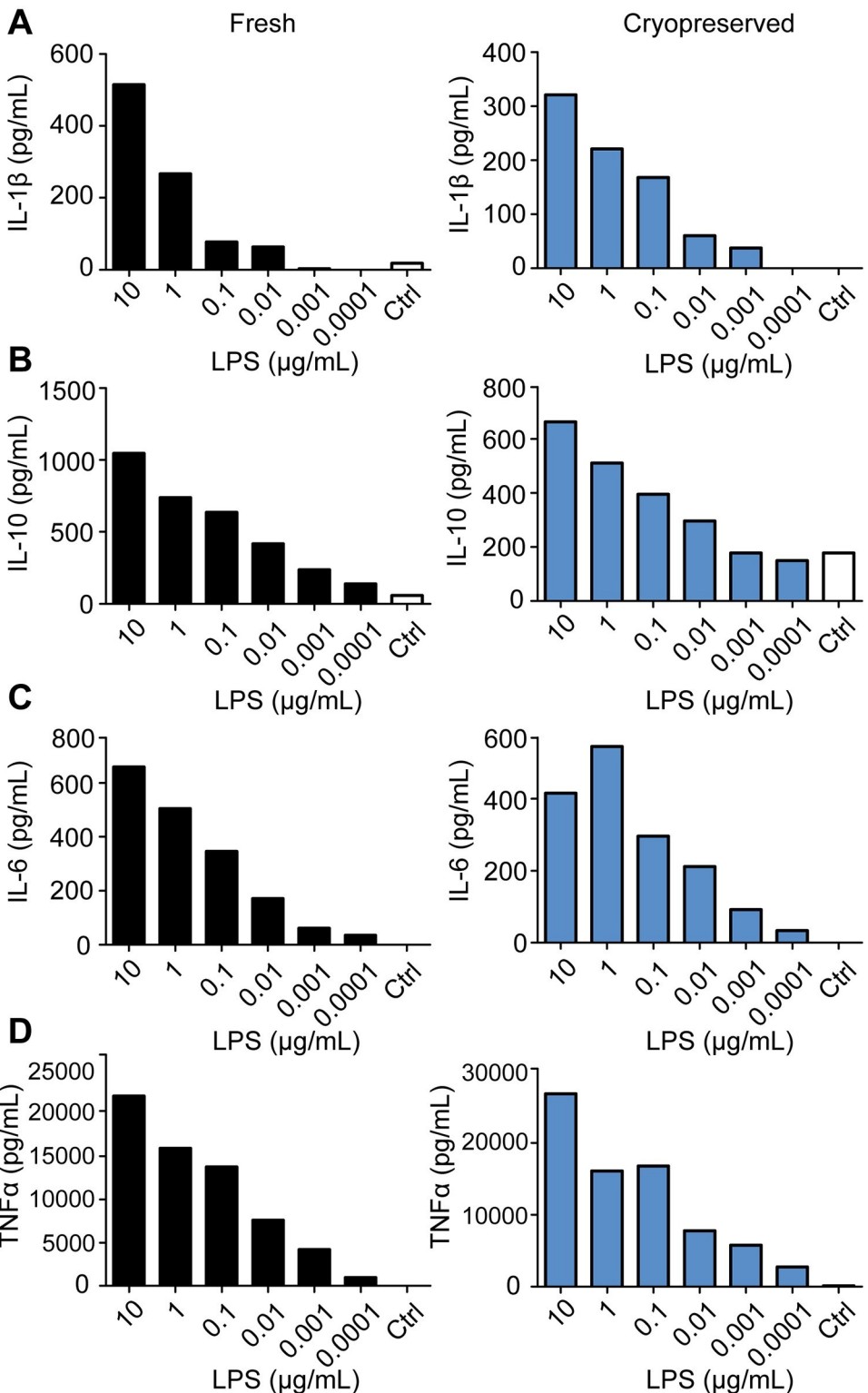

**Fig 4. FrhBMDCs and cryoBMDCs are able to produce cytokines in a dose-dependent manner upon LPS stimulation.** Immature frhBMDCs and cryoBMDCs (obtained from the same animal, n = 1) were stimulated with LPS or unstimulated using cell culture medium (negative control; Ctrl) for 24 hours. Levels of (A) IL-1β, (B) IL-10, (C) IL-6 and (D) TNFα were measured by ProcartaPlex. Note the different scales for cytokine production on the y-axis.

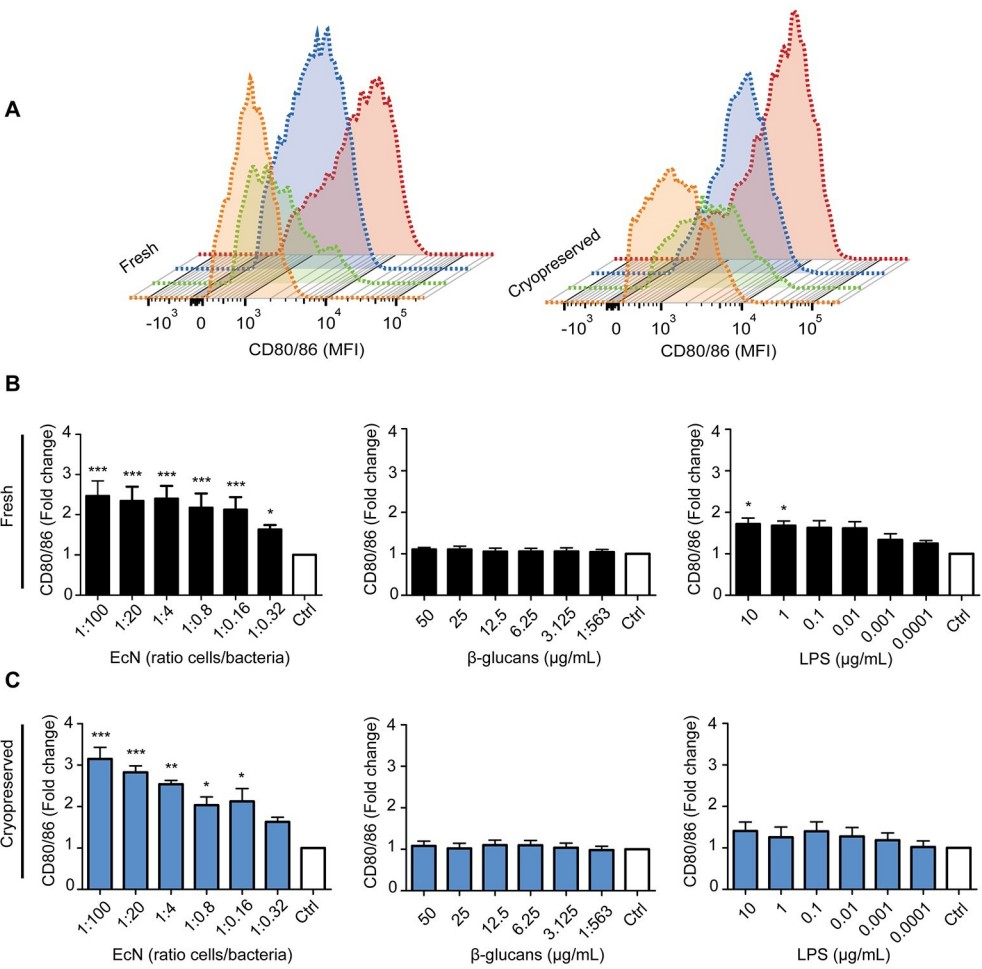

**Fig 5. *E. coli* Nissle 1917 (EcN), but not β-glucans, efficiently enhances the expression of CD80/86.** (A) Expression levels of CD80/86 on immature frhBMDCs and cryoBMDCs upon stimulation with the highest concentration of *E. coli* Nissle 1917 (red), β-glucans (green) and LPS (blue). Unstimulated cells using cell culture medium are represented by the orange histogram (negative control; Ctrl). The histograms are represented in a staggered offset format. (B) Immature frhBMDCs and (C) cryoBMDCs (obtained from the same animal) were stimulated with different concentrations of *E. coli* Nissle 1917, β-glucans or LPS. Unstimulated cells are represented by the white bars (negative control; Ctrl). After 24 hours, the upregulation of the maturation marker CD80/86 was measured using flow cytometry (n = 4 animals). Note the different scales for cytokine production on the y-axis. Data is presented as fold change to compensate for individual variation by dividing the MFI of stimulated BMDCs with the MFI of unstimulated BMDCs (Ctrl) of each animal. The data are shown as the means ± the standard error of the mean (SEM) of 4 animals. A one-way ANOVA with a Dunnett's post hoc test was performed, comparing multiple groups to the untreated cells (control): *** = $P<0.001$, **$P<0.01$ and * $P<0.05$.

Gibco™, MA, USA) without additional supplements, hereinafter referred to as plain cell culture medium. After washing, bacteria were resuspended at $1 \cdot 10^9$ colony forming units (CFU)/mL in plain cell culture medium and added in different concentrations to a 96-well cell culture plate as previously described. After 1 hour of incubation at 37°C, the 96-well plate was centrifuged at 300 g for 3 minutes and washed with plain cell culture medium. Subsequently, plain cell culture medium containing 100 µg/mL Gentamicin (Sigma-Aldrich, St. Louis, MO, USA) was added and incubated for 1 hour at 37°C to kill extracellular bacteria. After another washing step, cell culture medium containing 50 µg/mL Gentamicin was added to prevent bacterial growth during overnight incubation (24 hours). Porcine BMDCs were stimulated with either

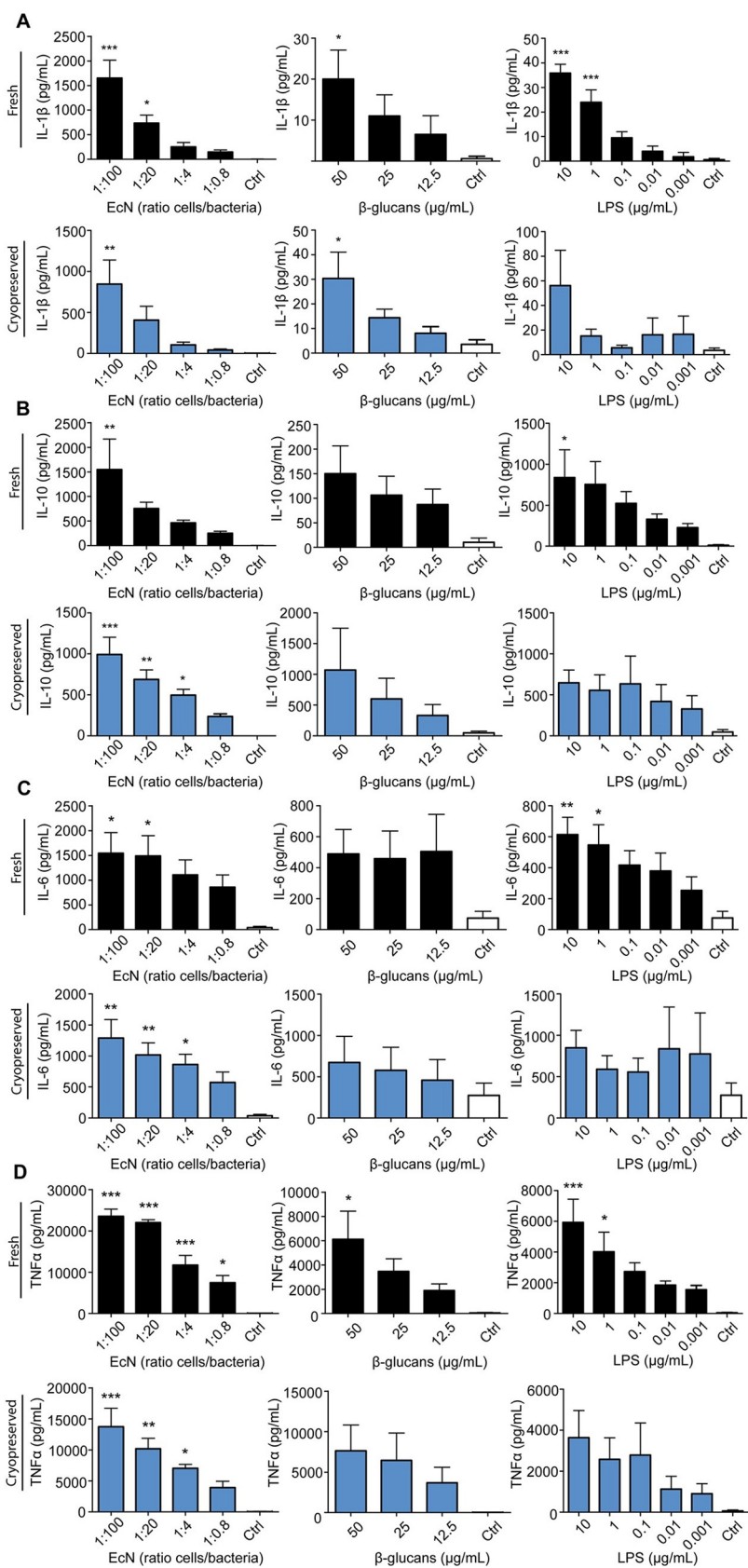

**Fig 6. *E. coli* Nissle 1917 (EcN) and β-glucans are able to induce a dose-dependent cytokine production.** Immature frhBMDCs (black) and cryoBMDCs (blue) (obtained from the same animal) were stimulated with EcN, β-glucans or LPS, or unstimulated using cell culture medium (negative control; Ctrl) for 24 hours. Levels of (A) IL-1β, (B) IL-10, (C) IL-6 and (D) TNFα were measured by ProcartaPlex (n = 4 animals). Note the different scales for cytokine production on the Y-axis. The data are shown as the means ± the standard error of the mean (SEM) of 4 animals. A one-way ANOVA with a Dunnett's post hoc test was performed, comparing multiple groups to the untreated cells (control): *** = P<0.001, **P<0.01 and * P<0.05.

LPS, β-glucans, EcN or cell culture medium (negative control) as described in section 'Generation and stimulation of BMDCs'. Cell culture supernatants were collected after 24 hours and stored at -80˚C until measurement.

## Flow cytometry

To assess the expression of cell maturation markers upon stimulation with LPS, β-glucans or EcN, wells with cultured BMDCs were incubated in FACS buffer ($Mg^{2+}$ and $Ca^{2+}$ free PBS (Lonza, Basel, Switzerland) containing 0.5% BSA fraction V (Roche, Basel, Switzerland), 2.0 mM EDTA (Merck, Kenalworth, NJ, USA) and FCS (Gibco™) at 37˚C for 30 minutes to facilitate the detachment of DCs. Surface marker expression was determined using fluorochrome-conjugated monoclonal antibodies specific for CD172a (PE Mouse Anti-Pig Monocyte/Granulocyte, clone 74-22-15 BD Biosciences Cat# 561499, RRID:AB_10694894) diluted 1:320 in FACS buffer [22], SLA Class-II (porcine equivalent of MHC-II) (SLA Class-II DR: FITC, clone 2E9/13, Thermo Fisher Scientific Cat# MA5-16490, RRID:AB_2538001) diluted 1:160 in FACS buffer [23] and APC conjugated human CD152 (CTLA-4) murine Ig fusion protein diluted 1:320 in FACS buffer (Ancell, Bayport, MN) to detect CD80 and CD86 [24–26]. Before measuring, DNA-binding 7-AAD (BD Bioscience) was added to discriminate viable from dead cells. Matching fluorescent compensation controls Rat IgG2b κ FITC (eB149/10H5, eBioscience™, Rat IgG2a κ PE (RTK2758, BioLegend) and Armenian Hamster IgG APC (HTK888, BioLegend) were included by adding the–to the specific fluorochrome corresponding–compensation Abs to compensation beads (Ultracomp eBeads™, ThermoFisher Scientific™). In addition, unstained BMDCs were measured to detect autofluorescence. Thereafter, cells were resuspended in 100μl FACS buffer (1:50) and DNA-binding 7-AAD (BD Bioscience) was added to discriminate viable from dead cells. Subsequently, all cell suspensions were acquired (average of 10.000 events) on a BD FACS Canto II (BD Biosciences) and analysed using the FlowJo™ software (BD-Biosciences). Median Fluorescence Intensity (MFI) values were calculated for the CD172a+/high CD80/86+ and SLA Class-II+ population (Fig 2, panel 6) to determine the expression of SLA Class-II and CD80/86 proteins on stimulated BMDCs relative to unstimulated BMDCs (negative control). Phenotypic identification of DCs by flow cytometry was initially performed by using forward scatter (cell size) (to exclude debris and red blood cells), side scatter (cell granularity) and by identifying single- and viable cells. In accordance with previous studies, DCs were further identified by high expression levels of the myeloid cell marker CD172a (CD172a+/high) and high expression levels of CD80/86+ and SLA Class-II+ (Fig 2 and S2 Fig) [5, 25]. Cells expressing intermediate levels of CD172a (CD172a+/medium) were excluded from the analysis based on the lack of CD80/86 and SLA Class-II co-expression (S3 Fig). This lack of CD80/86 and SLA Class-II co-expression was even observed after LPS stimulation [27]).

## Cytokine measurement

Porcine BMDCs were stimulated with either LPS, β-glucans or EcN as mentioned above. Subsequently, cell culture supernatants were collected after 24 hours and stored at -80˚C until

measurement. The levels of different cytokines (IFN-α, IFN-γ, IL-1β, IL-4, IL-6, IL-8, IL-10, IL-12p40 and TNFα) were measured in the supernatants of BMDC cultures (pg/mL) by using the bead-based Porcine Procartaplex assay™ according to the manufacturer's recommended protocols (ThermoFisher Scientific™) and read on a BioPlex® machine (Bio-Rad). Best-fit standard curves for each analyte were established by Bio-plex manager software (Version 3).

### Statistical analysis

Statistical differences were tested by a one-way analysis of variance (ANOVA) with a Dunnett's post hoc test, comparing multiple groups to the untreated cells (control). Data represented as fold-change was transformed for statistical analysis using the logarithm of the raw data. A P-value of 0.05 was considered statistically significant. Data is represented as mean ± standard error of the mean (SEM). Significant differences were indicated by asterisks, respectively; *** = $P<0.001$, **$P<0.01$ and * $P<0.05$. GraphPad Prism 5 software (GraphPad, US) was used for all statistical analyses. Clustering of immunological data (surface marker expression and cytokine production) and stimulation conditions of frhBMDC and cryoBMDC was done by principal component analysis (PCA) using R statistical software (Version 3.5.2).

## Results

### 1. Generation of porcine BMDCs

Stimulation of bone marrow cells with recombinant porcine GM-CSF (rpGM-CSF) generated cells with dendritic cell like morphology (Fig 1) characterized by dendrites; membrane extensions that could be observed from day 3 onwards [27]. Cells which were highly positive for the myeloid cell marker CD172a (CD172a$^{+/high}$) expressed relatively high levels of CD80/86 and SLA Class-II (Fig 2). Cells expressing intermediate levels of CD172a (CD172a$^{+/medium}$) expressed low levels of CD80/86 and SLA Class-II co-expression and were excluded from the analysis (S3 Fig). Cells cultured from fresh bone marrow cells contained on average lower percentages of CD172a$^{+/high}$ cells in comparison to cells cultured from cryopreserved bone marrow cells (24% and 62%, respectively). The percentage of viable cells was similar between the frhBMDCs and cryoBMDCs cultures (>90% viable cells).

### 2. Porcine BMDCs upregulate the maturation markers CD80/86 and produce several DC signature cytokines upon stimulation with LPS

After generation and phenotyping of porcine BMDCs, their ability to upregulate maturation markers (CD80/86 and SLA Class-II) as well as cytokine production were determined. Upon stimulation with LPS, frhBMDCs and cryoBMDCs showed increased expression (MFI) of the co-stimulatory marker CD80/86 in a dose-dependent manner (Fig 3). Despite differences in the basal CD80/86 expression levels between frhBMDCs and cryoBMDCs (MFI of 753 and 1165, respectively), their ability to upregulate CD80/86 upon stimulation with LPS (e.g. 10 μg/mL) is similar (stimulation index (SI) of 3.6 and 4.3, respectively). In addition, LPS induced SLA Class-II upregulation (S4 Fig), but this was less consistent between different concentrations and experiments in comparison to CD80/86. The percentages of CD80/86 positive or SLA Class-II positive cells remained constant upon stimulation [27]. Subsequently, the production of several cytokines secreted by BMDC upon stimulation was analysed, including IL-1β, IL-6, IL-10, TNFα, IL-4, IL-8, IL-12p40, IFN-α and IFN-γ. Similar to the expression of CD80/86, levels of the cytokines IL-1β, IL-6, IL-10, and TNFα increased upon LPS stimulation in a dose-dependent manner (Fig 4). No or inconsistent effects of LPS were observed for the induction of cytokines IL-4, IL-8, IL-12p40, IFN-α and IFN-γ [27]. Moreover, similar levels of

cytokines were produced by cryoBMDCs upon LPS stimulation. However, LPS stimulated cryoBMDCs produced lower amounts of IL-10 and higher amounts of TNFα in comparison to frhBMDCs.

### 3. *E. coli* Nissle 1917, but not β-glucans, efficiently enhanced the expression of the maturation marker CD80/86 on frhBMDCs and cryoBMDCs

Subsequent to the analysis of LPS stimulated BMDCs (control), two feed additives (β-glucans and EcN) were tested to assess their immunomodulatory properties *in vitro* on frhBMDCs and cryoBMDCs. Upon stimulation with EcN, but not β-glucans, the maturation marker CD80/86 was clearly upregulated on both types of BMDCs (Fig 5A–5C). These results also demonstrate that less than one bacterium per DC was required for significant upregulation of CD80/86 in both frhBMDCs and cryoBMDCs. Interestingly, these relatively low numbers of EcN bacteria elicited a much stronger response compared to the maximum response that could be obtained with LPS. Despite the dose-dependent upregulation of CD80/86 on LPS stimulated frhBMDCs, LPS induced upregulation of CD80/86 on cryoBMDCs did not reach significance. Unexpectedly, SLA Class-II was neither upregulated by the feed additives nor by the control (LPS) (S5 Fig). Unstimulated frhBMDCs and cryoBMDCs showed similar median fluorescence intensities assessing the co-stimulatory marker CD80/86 (MFI of 1600 and 2200, respectively). Furthermore, EcN stimulated frhBMDCs and cryoBMDCs showed similar basal levels of CD80/86 expressed in median fluorescent intensity (MFI of 1807 and 1581, respectively). However, EcN stimulated cryoBMDCs showed a higher SI in comparison to EcN stimulated frhBMDCs (SI of 2.3 and 3.1, respectively).

### 4. FrhBMDCs and cryoBMDCs produce cytokines upon stimulation with *E. coli* Nissle 1917 and β-glucans

Subsequent to the analysis of surface expression of BMDC maturation markers upon stimulation with EcN and β-glucans, cell culture supernatant was collected and tested for the presence of the cytokines IL-1β, IL-6, IL-10 and TNFα. A clear dose-dependent induction of all cytokines was seen for both β-glucans and EcN (Figs 5 and 6). In comparison to β-glucans and LPS, EcN induced much stronger cytokine responses in frhBMDCs and cryoBMDCs. Despite the fact that both types of BMDCs responded in a similar fashion, cryopreservation affected the amount of cytokines produced by these cells. In general, cryopreserved BMDCs were less able to produce cytokines upon stimulation with EcN but produced more cytokines when stimulated with β-glucans. For example, cryopreservation resulted in an impaired IL-10 response by BMDC upon stimulation with EcN (Fig 5C). In contrast, β-glucans was shown to enhance the IL-10 production of BMDC from cryopreserved origin.

### 5. Clustering by principal component analysis

To investigate and visualize the variation and patterns of our dataset in a single figure, results obtained from the various immunological analyses were used to conduct a two-dimensional principal component analysis (PCA) (Fig 7). The PCA indicates the amount of variation retained by each principal component (PC1 scores on the x-axis and PC2 scores on the y-axis). For the frhBMDC culture, the amount of variation explained by PC1 and PC2 is 66.5% and 13.3% respectively. For the cryoBMDC culture, the amount of variation explained by PC1 and PC2 is 39.1% and 22.2% respectively. In other words, 79.8% (in frhBMDC cultures) and 61.3% (in cryoBMDC cultures) of all variance of immune responses *in vitro* could be explained by only 2 principal components. Fig 7 includes all *in vitro* measured immune parameters (CD80/

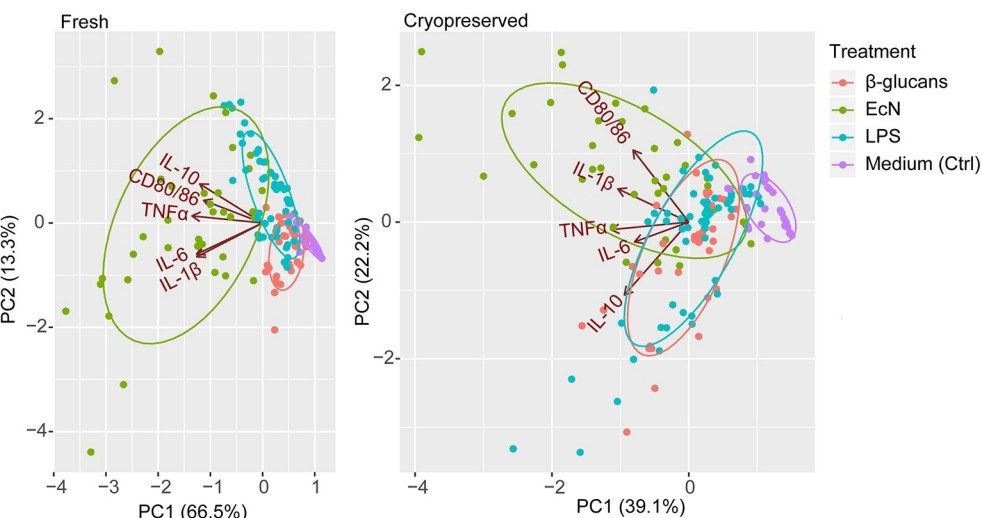

**Fig 7. PCA grouped by stimulation conditions of frhBMDC and cryoBMDC cultures.** The PCA includes data from different animals (n = 4), different treatment concentrations and technical replicates per stimulation condition (enveloped) that were analysed by flow cytometry and cytokine analysis. Each dot represents a single measurement. The principle components (PC1 and PC2) indicate the amount of variation which can be explained by the model. The PCA plots exhibit the measured immunological parameters, indicated by arrows (CD80/86 expression and cytokine levels; IL-1β, IL-6, IL-10 and TNFα). The length of the arrows approximates the variance of the variables, whereas the angels between them approximate their correlations.

86 expression and IL-1β, IL-6, IL-10 and TNFα levels) from stimulated frhBMDCs and cryoBMDCs. Data points (single measurements) are enveloped by treatment. For both frhBMDC and cryoBMDC, a distinct grouping of immune parameters can be observed for each stimulus. Most of these groups overlap, but consistently the enveloped data points of the EcN group stands out for both frhBMDCs and cryoBMDCs.

## Discussion

Immunomodulation by feed additives offers suitable options to enhance immune competence, thereby contributing to the maintenance of animal health and well-being. Feed additives that possess immune stimulating activity could induce maturation of cells of the immune system, resulting in a more efficient response to microbial infections. Mature APCs express high levels of the co-stimulatory molecules CD80/86 and are therefore more likely to reach the threshold for T-cell activation. In addition, the production of cytokines by these APCs can also contribute to the development of a more effective immune response against the presented antigen. To date, the effects of feed additives on the immune system are not completely understood and tools to assess these effects in pigs are limited. Especially DCs are of interest, as they are known to interact directly with antigens present in the intestinal luminal content, thereby directing adaptive immune responses [2]. These intestinal DCs are considered to be the target for β-glucan and probiotic activity but are not easily accessible for *in vitro* studies. However, DCs can be generated *in vitro* from more accessible precursor cells, like blood derived monocytes or bone marrow cells [25]. In this study, cultured BMDCs were used to investigate the immunomodulatory effects of β-glucans and EcN *in vitro*, using the expression of DC maturation markers (SLA Class-II and CD80/86) and production of cytokines as read-out parameters. We observed a dose-dependent production of the cytokines TNFα, IL-1β, IL-6 and IL-10 upon stimulation with β-glucans and EcN. In comparison to other *in vitro* studies using DCs, a similar spectrum of cytokines was produced upon stimulation [7, 17]. In contrast, Sonck et al.

detected a much higher amount of IL-1β and lower amounts of IL-10 after stimulation of porcine MoDCs with yeast-derived β-glucans, indicating a different responsiveness of both types of DCs. This is supported by the observation that yeast-derived β-glucans did not upregulate the maturation marker CD80/86 on our BMDCs, while it did on porcine MoDCs. On the other hand, these porcine MoDC did not upregulate CD80/86 after stimulation with LPS, while these markers were upregulated in a dose-dependent manner by this stimulus in our BMDCs (Fig 5) [7]. Moreover, the feed additives were not able to enhance SLA Class-II (S5 Fig). A comparison between porcine BMDC and porcine MoDC demonstrated differences in their SLA Class-II expression in which, in line with our results, porcine BMDCs were not able to upregulate SLA Class-II upon stimulation with LPS [5]. A possible explanation for the inability of β-glucans to enhance DC maturation marker expression on BMDCs is the absence of specific β-glucans receptors such as Dectin-1, as it is currently unknown which cell types are expressing Dectin-1. The primary receptor for β-glucans, Dectin-1, is present in porcine intestinal tissues and on porcine MoDCs, but this has not been examined for porcine BMDCs [7, 28]. The cytokine production could also be a consequence β-glucan binding to other potential receptors, such as complement receptor 3 (CR3), lactosylceramide and scavenger receptors [29].

Previous studies used porcine Monocyte-derived Dendritic Cells (MoDC) to study *in vitro* effects of immunomodulatory feed additives [7, 30], as MoDC cultures are well established and contain a relatively homogenous population of DCs, whereas BMDC cultures are heterogeneous [31]. However, MoDC cultures require fresh monocyte isolation from blood, as monocytes are known to be sensitive to freezing [32]. This will introduce time-to-time and animal-to-animal variation which may greatly impair reproducible comparison of immunomodulatory feed additives tested *in vitro* at different time points. A large stock of cryopreserved aliquots of bone marrow cells from a single animal would greatly facilitate such *in vitro* research. Human MoDCs differentiated from cryopreserved monocytes expressed lower levels of maturation markers (CD40, CD80, CD83 and CD86) and showed to have an altered cytokine response upon stimulation [33, 34] In contrast, Hayden and co-workers demonstrated that the properties of MoDCs derived from cryopreserved monocytes were equivalent to MoDCs generated from freshly isolated human monocytes [35]. In this study, frhBMDCs were compared to cryoBMDCs in their ability to upregulate maturation markers and produce cytokines upon stimulation with different feed additives which use different signalling pathways, including yeast-derived β-glucans and the probiotic strain *E. coli* Nissle 1917 (EcN). We demonstrated that cryopreservation of bone marrow cells resulted in elevated expression levels of the maturation marker CD80/86 on differentiated BMDCs and somewhat increased the production of IL-6 and IL-10 by unstimulated DCs. This indicates that cryopreservation of bone marrow cells leads to partially maturated BMDCs, characterised by enhanced expression of cell maturation markers and altered cytokine production by unstimulated cells as also observed in other studies [36, 37]. Cryopreservation also resulted in an increased animal-to-animal variation (Fig 6). However, cryopreservation did not have profound effects on the dose-dependent induction of CD80/86 nor on the production of cytokines upon stimulation with LPS, β-glucans and EcN (Figs 4 and 6), making these cells also suitable for the assessment of the immunomodulatory effects of feed compounds.

The PCA (Fig 7) demonstrates the overlap between the immune responses to LPS and β-glucans exposure, while stimulation with EcN reveals clearly distinct grouping or responsiveness. This is in line with our previous results (Figs 5 and 6), in which EcN showed to be the most potent immunostimulator, inducing a much stronger upregulation of CD80/86 and production of cytokines. These results suggest that stimulation with EcN contributes other (stimulatory) factors in comparison to LPS or β-glucans. The relatively strong response to EcN in

comparison to LPS could be explained by the fact that EcN has a greater number of microbe associated molecular patterns (MAMPs) that can bind to pattern recognition receptors (PRRs) on DC [38]. Moreover, cryoBMDCs stimulated with EcN resulted in a homogeneous and separate cluster from the other conditions, albeit these stimulation conditions resulted in more variation compared to frhBMDCs. Fig 7 also demonstrates that all *in vitro* immune responses show a clear positive correlation upon stimulation (all arrows point in the same direction) and each response contributes equally to this variation (arrows are of equal length). For cryoBMDCs, the CD80/86 expression and production of IL-10 cluster appeared more independently from each other as compared to the other cytokines while retaining their positive correlation (arrows pointing in the same direction but at a right angle). In addition, the TNFα pro-inflammatory response appears to be independently regulated from the anti-inflammatory IL-10 response, although both responses were shown to be correlated in other literature [39].

In summary, this study demonstrates that yeast-derived β-glucans (MacroGard®) and EcN are both able to enhance DC maturation, but in a differential manner. A more mature DC phenotype could contribute to a more efficient response to infections. However, *in vivo* studies are required to verify these results. In addition, this study shows that BMDCs originating from fresh or cryopreserved cultured porcine bone marrow cells can both be used as an *in vitro* pre-screening tool to allow a more evidence-based pre-selection of promising immune modulators for validation in *in vivo* studies in pigs.

## Supporting information

**S1 Fig. β-glucans does not contain LPS and does not induce hTLR4 mediated NF-κB activation.** (**A**) 100 μg/mL β-glucans (MacroGard®) was tested for LPS contamination using a recombinant factor C LAL assay preparation. A 5 EU spiked control was included (n = 1). (**B**) Different concentrations (1 mg/mL– 0.1 μg/mL) of commercial β-glucans (MacroGard®) and LPS (100 pg/mL) were tested for their NF-κB activation via hTLR4 (n = 3).
(TIF)

**S2 Fig. Phenotype of cryopreserved cultured porcine mononuclear phagocytes.** Gating strategy following multicolour flow cytometry staining using Abs against CD172a, SLA Class-II and CD80/86. Cells showing high forward scatter (FSC-A) and side scatter (SSC-A) profiles were gated, followed by the selection of single cells (FSC-W/H and SSC-W/H) and viable cells (SSC-A/7-AAD). Among these cells, BMDCs were defined as the CD172a+/high cells (SSC-A/ CD172a) expressing SLA Class-II and CD80/86.
(TIF)

**S3 Fig. Phenotype of CD172a+/- (intermediate) cell population.** Gating strategy of (A) frhBMDCs and (B) cryoBMDCs following multicolour flow cytometry staining using Abs against CD172a, SLA Class-II and CD80/86. The CD172a+/- (intermediate) cell population (SSC-A/CD172a) does not express SLA Class-II and CD80/86 in both frhBMDC and cryoBMDC cell cultures.
(TIF)

**S4 Fig. FrhBMDCs and cryoBMDCs upregulate SLA Class-II in a dose-dependent manner upon stimulation with LPS.** (A) FrhBMDCs and cryoBMDCs (obtained from the same animal, n = 1) were stimulated with different concentrations of LPS or unstimulated using cell culture medium (negative control; Ctrl). After 24 hours, the expression (MFI) of the maturation markers SLA Class-II were measured using Flow Cytometry. The data are shown as the means ± the standard error of the mean (SEM) of three technical replicates. A one-way ANOVA with a Dunnett's post hoc test was performed, comparing multiple groups to the

untreated cells (control): *** = P<0.001, **P<0.01 and * P<0.05. (B) Representative contour plots of SLA Class-II expression on LPS stimulated frhBMDCs and cryoBMDCs. The contour plots are based on forward scatter (y-axis) and SLA Class-II expression (x-axis). The highest concentration of LPS (10 μg/mL) and cell culture medium (negative control; blue) are presented in this figure.
(TIF)

**S5 Fig. SLA Class-II is not upregulated upon stimulation with EcN, β-glucans or LPS.**
Immature (A) frhBMDCs and (B) cryoBMDCs (obtained from the same animal) were stimulated with different concentrations of *E. coli* Nissle 1917, β-glucans or LPS. Unstimulated cells are represented by the white bars (negative control; Ctrl). After 24 hours, the upregulation of SLA Class-II was measured using Flow Cytometry (n = 4 animals). Relative fold change was calculated by dividing the MFI of stimulated BMDC/MFI of unstimulated BMDC (Ctrl) of each animal. The data are shown as the means ± the standard error of the mean (SEM) of 4 animals. A one-way ANOVA with a Dunnett's post hoc test was performed, comparing multiple groups to the untreated cells (control): *** = P<0.001, **P<0.01 and * P<0.05.
(TIF)

## Acknowledgments

We would like to thank H.J.A. (Hugo) de Vries (WUR) and Dr. ir. H.K. (Henk) Parmentier (WUR) for their valuable contribution to the PCA plots.

## Author Contributions

**Conceptualization:** Mirelle Geervliet, Christine A. Jansen, Victor P. M. G. Rutten, Huub F. J. Savelkoul, Edwin Tijhaar.

**Data curation:** Mirelle Geervliet, Laura C. P. Lute.

**Formal analysis:** Mirelle Geervliet, Laura C. P. Lute.

**Funding acquisition:** Huub F. J. Savelkoul.

**Investigation:** Mirelle Geervliet, Laura C. P. Lute, Edwin Tijhaar.

**Methodology:** Mirelle Geervliet, Laura C. P. Lute, Edwin Tijhaar.

**Project administration:** Huub F. J. Savelkoul, Edwin Tijhaar.

**Supervision:** Huub F. J. Savelkoul, Edwin Tijhaar.

**Validation:** Mirelle Geervliet, Huub F. J. Savelkoul, Edwin Tijhaar.

**Visualization:** Mirelle Geervliet.

**Writing – original draft:** Mirelle Geervliet.

**Writing – review & editing:** Mirelle Geervliet, Christine A. Jansen, Victor P. M. G. Rutten, Huub F. J. Savelkoul, Edwin Tijhaar.

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
