## [Decision Letter · Decision Letter 0]

18 Feb 2020

PONE-D-19-28012

Differential immunomodulation of porcine bone marrow derived dendritic cells by E. coli Nissle 1917 and β-glucans

PLOS ONE

Dear Mrs Geervliet,

Thank you for submitting your manuscript to PLOS ONE. After careful consideration, we feel that it has merit but does not fully meet PLOS ONE’s publication criteria as it currently stands. Therefore, we invite you to submit a revised version of the manuscript that addresses the points raised during the review process.

I would like to sincerely apologise for the delay you have incurred with your submission. It has been exceptionally difficult to secure reviewers to evaluate your study. We have now received three completed reviews; their comments are available below.

Reviewer#1 and #3 have raised some concerns about the study that need to be addressed in a revision. Both reviewers have noted that a clearer rationale for your study should be provided. Please revise the manuscript to address all the reviewer's comments in a point-by-point response in order to ensure it is meeting the journal's publication criteria. Please note that the revised manuscript will need to undergo further review, we thus cannot at this point anticipate the outcome of the evaluation process.

We would appreciate receiving your revised manuscript by Apr 02 2020 11:59PM. To enhance the reproducibility of your results, we recommend that if applicable you deposit your laboratory protocols in protocols.io, where a protocol can be assigned its own identifier (DOI) such that it can be cited independently in the future. For instructions see: http://journals.plos.org/plosone/s/submission-guidelines#loc-laboratory-protocols

We look forward to receiving your revised manuscript.

Kind regards,

Miquel Vall-llosera Camps

Associate Editor

PLOS ONE

Journal Requirements:

2. To comply with PLOS ONE submissions requirements, please provide methods of sacrifice in the Methods section of your manuscript.

Reviewers' comments:

Reviewer's Responses to Questions

**Comments to the Author**

1. Is the manuscript technically sound, and do the data support the conclusions?

Reviewer #1: Yes

Reviewer #2: Yes

Reviewer #3: Yes

2. Has the statistical analysis been performed appropriately and rigorously? 

Reviewer #1: I Don't Know

Reviewer #2: Yes

Reviewer #3: Yes

3. Have the authors made all data underlying the findings in their manuscript fully available?

Reviewer #1: Yes

Reviewer #2: Yes

Reviewer #3: Yes

4. Is the manuscript presented in an intelligible fashion and written in standard English?

Reviewer #1: Yes

Reviewer #2: Yes

Reviewer #3: Yes

5. Review Comments to the Author

Reviewer #1: Authors describe an experiment designed study the possible immunomodulating effects of EcN and betaglucans on dendritic cells. It is in principle a study worth publishing with interesting results. There are however a few points to be resolved before publication. In the introduction it remains unclear what exactly is meant by immunomodulation and what the authors expect from it in terms of animal health. As is, it is very vague. For instance, do the authors aim for higher titers, stronger inflammatory responses etc., and would that lead to better health?

Also, the results obtained should be discussed in the light of the above. It is currently not clear from the discussion what the results actually mean for the animals.

The other important point is that the information about betaglucans is too summary. There are many different types and sources of betaglucans, and different properties and immunomodulating effects have been ascribed to them e.g. soluble vs particulate etc. This should be at least briefly be mentioned in the introduction. Furthermore, it is not described what type of betaglucan was used (soluble? DP?) and what the other 40% consists of.

Reviewer #2: I this study, authors assess the immunomodulatory potential of E. coli Nissle 1917 and β-glucans in vitro using porcine bone marrow-derived dendritic cell. The finding of this study are interesting, demonstrated that both β-glucans and E. coli Nissle 1917 were able to enhance dendritic cell maturation, but in a differential manner. In my opinion the manuscript is well structured and well written with minor topographical errors. I recommend to accept this paper.

Reviewer #3: This is a methodologically sound study describing the effects of foodstuffs, based on a strain of E. coli and Beta-glucans, for their effects on dendritic cells (DC), as modeled by DC derived in vitro from bone marrow cells. I presume that it the commercial use of these particular materials that has led to their selection for this study but this isn't very explicit. There is a sound rationale for focusing on DC in this sort of analysis but a major limitation of this sort of study is that it is unclear what sort of DC is relevant, what defines a 'good' response in DC and what are the circumstances in which these responses can be induced in the intestine in vivo. Perhaps these issues could be touched upon the manuscript.

I would make the following addition minor comments and suggestions:

- What was the control for the CD152-Ig staining reagent?

- How Were MFI values calculated ? Was the MFI for the control staining subtracted?

- It would be good to show example FACS plots for the relevant markers in stimulated cells in the main body of the paper (so the reader can get an idea of how the MFI values shown in graphs were derived.

- Figure legends should consistently state the number of replicates/independent experiments. It is recognised that the nature of work on pigs mean that replication will be limited - but it needs to be explicit.

- There are some minor typos etc:'rpm' instead of 'g' (L153); lack of superscript (L119); Figure S4 is not referred to in the text; the figure panels in S2 are not explained in the legend; what is FCM (L307)?

- Clarify what 'remains to be detected' (L399) means? There is no data? Ambiguous data? Conflicting data?

- What does 'appeared' (L266) mean?

- The values of the PCA analysis is unclear. It isn't really explained or described in the Results section ( there is some attempt to address this belatedly in Discussion) and it feels a bit of an afterthought.

6. PLOS authors have the option to publish the peer review history of their article (what does this mean?). If published, this will include your full peer review and any attached files.

Reviewer #1: No

Reviewer #2: No

Reviewer #3: No

---

## [Author Response · Author response to Decision Letter 0]

2 Apr 2020

Dear editor,

We would like to thank you and the reviewers for the valuable comments on our manuscript entitled “Differential immunomodulation of porcine bone marrow derived dendritic cells by E. coli Nissle 1917 and β-glucans” by Mirelle Geervliet1, Laura C.P. Lute1, Christine A. Jansen2, Victor P.M.G. Rutten2,3, Huub F.J. Savelkoul1, Edwin Tijhaar1*.

We have carefully read the comments that were presented and we resubmitted our paper including revisions. Please note that the line numbers (in yellow) correspond to the line numbers in the revised document.

Below we give a detailed point-by-point reply regarding the comments and we adapted the manuscript accordingly.

ACADEMIC EDITOR SPECIFIC COMMENTS:

1. Please ensure that your manuscript meets PLOS ONE's style requirements, including those for file naming. The PLOS ONE style templates can be found at:

- http://www.journals.plos.org/plosone/s/file?id=wjVg/PLOSOne_formatting_sample_main_body.pdf

- http://www.journals.plos.org/plosone/s/file?id=ba62/PLOSOne_formatting_sample_title_authors_affiliations.pdf

Authors response: we have adopted the requirements according to your suggestions in the revised manuscript. The file names are adjusted according to the PLOSONE requirements.

2. To comply with PLOS ONE submissions requirements, please provide methods of sacrifice in the Methods section of your manuscript.

Authors response: The methods of sacrifice is now included in the Methods section (Ethics statement) of the manuscript (Lines 129 – 130).

Authors response: We removed the phrase ‘data not shown’ from the manuscript and referred to the public repository accordingly. 

Authors response: Thank you for this information. We have now obtained the requested repository information and DOI number. We removed the phrase ‘data not shown’ from the manuscript and referred to the public repository in lines 223, 254, 257 and 288. Therefore no changes regarding our data availability statement. 

REVIEWER COMMENTS:

Reviewer #1: Authors describe an experiment designed study the possible immunomodulating effects of EcN and betaglucans on dendritic cells. It is in principle a study worth publishing with interesting results. There are however a few points to be resolved before publication. 

• In the introduction it remains unclear what exactly is meant by immunomodulation and what the authors expect from it in terms of animal health. As is, it is very vague. For instance, do the authors aim for higher titers, stronger inflammatory responses etc., and would that lead to better health?

Authors response: We reformulated the text in the introduction (Lines 50 - 59) and discussion (Lines 404 - 409 and Lines 481 - 483) to make clear what we mean with ‘immunomodulation’; a more primed/mature state of the immune system that may contribute to a better resistance against (bacterial and viral) infections. In addition, we explain how maturation of DCs could benefit this (Lines 59 - 61). What it could mean in terms of animal health is addressed at the end of the abstract (Lines 38 - 39) and at the end of the discussion (Lines 481 - 483). 

• Also, the results obtained should be discussed in the light of the above. It is currently not clear from the discussion what the results actually mean for the animals.

Authors response: As mentioned above, at the end of the abstract (Lines 38 - 40) and at the end of the discussion (Lines 481 - 483) it is better explained what the results of this study could mean for animal health.

• The other important point is that the information about betaglucans is too summary. There are many different types and sources of betaglucans, and different properties and immunomodulating effects have been ascribed to them e.g. soluble vs particulate etc. This should be at least briefly be mentioned in the introduction. 

Authors response: This is indeed important to address in this article. We added additional lines about the different types and properties of betaglucans (with references) in the introduction (Lines 77 - 87).

• Furthermore, it is not described what type of betaglucan was used (soluble? DP?) and what the other 40% consists of.

Authors response: The type of betaglucans used (particulate/structure) and the composition of MacroGard® (percentage), including the non-betaglucan part, is now described in Lines 77 - 78 and Lines 160 - 163. 

Reviewer #3: This is a methodologically sound study describing the effects of foodstuffs, based on a strain of E. coli and Beta-glucans, for their effects on dendritic cells (DC), as modeled by DC derived in vitro from bone marrow cells. 

• I presume that it the commercial use of these particular materials that has led to their selection for this study but this isn't very explicit. 

Authors response: The promising immunomodulatory properties of MacroGard (yeast-derived and particulate beta-glucans) and its commercial use in pig feed has led to the selection of this type of beta-glucans. This is now clarified in the introduction with references (lines 81 – 87). EcN is known as a versatile probiotic interacting with cells of the immune system including antigen presenting cells such as DCs. Despite its (commercial) use over 100 years in humans there is only limited information of its immunomodulatory effect in pigs (Lines 87 – 93), prompting us to examine its effect on porcine DCs. 

• There is a sound rationale for focusing on DC in this sort of analysis but a major limitation of this sort of study is that it is unclear what sort of DC is relevant, what defines a 'good' response in DC and what are the circumstances in which these responses can be induced in the intestine in vivo. Perhaps these issues could be touched upon the manuscript.

Authors response: These are indeed valid points to add to the discussion. There are different types of DCs which possess different characteristic and functions based on their location and maturation status. Intestinal DCs are considered to be the target for β-glucan and probiotic activity in vivo, but are not easily accessible for in vitro studies. We have now added a more extensive explanation on this point (Lines 411 – 414). It is indeed difficult to define what is a ‘good’ DC response. Higher expression levels of the co-stimulatory molecules CD80/86 on antigen presenting cells implies that these cells are more likely to reach the required threshold for T-cell activation. In addition, the production of cytokines by the antigen presenting cells can also contribute to the development of a more effective immune response against the presented antigen (Lines 404 - 409). The circumstances in which these responses can be induced in the intestine in vivo is briefly mentioned in Lines 59 - 61 and Lines 411-412. We realize what limitations in vitro studies can have for the situation in vivo. We consider this in vitro model as a valuable method to pre-screen putative feed additives, and to select the more promising ones that will then have to be validated in vivo. This is now indicated in Lines 481 – 483. 

I would make the following addition minor comments and suggestions:

• What was the control for the CD152-Ig staining reagent?

Authors response: Thank you for this question. Different controls have been used in this study. Firstly, single color stains for setting the compensation have been described in the Materials and Methods section of the manuscript. Secondly, we used unstained cells to detect possible autofluorescence. We noticed that this control accidently was missing in the manuscript, but is now included in Line 208. Because of the limited number of colors used (FITC, PE and APC) with very different emission wavelengths we did not include FMO controls. Retrospectively, the FMO control could have been included. Nonetheless, we are confident about the specific staining of the CTLA4Ig Fusion Protein as it is widely used and accepted in in vitro and in vivo studies in pigs and our CD80/86 staining/expression is in line with what has been described in other literature. We have added now added additional references (see references (1-3) below). In addition, Figure S2 shows that cells that express SLA Class-II also express CD80/86 as expected. Cells that do not express SLA Class-II do not express CD80/86 and thus no unspecific binding occurs. 

• How were MFI values calculated? Was the MFI for the control staining subtracted?

Authors response: The MFI values were calculated for the CD172a+/high CD80/86+ and SLA Class-II+ population (Figure 2, panel 6). A new and more clear explanation is added in the Materials and Methods section (Lines 212 - 213). In addition, we adjusted the gating of the SLA Class-II/CD80/86 double positive population to make it more clear for the reader how the MFI was actually calculated (Figure 2, panel 6). The MFI of the control is not subtracted. The MFI of the control is indicated by the white bar in Figure 3, 4, 5 and 6 (and S4 and S5 Figure).

• It would be good to show example FACS plots for the relevant markers in stimulated cells in the main body of the paper (so the reader can get an idea of how the MFI values shown in graphs were derived. 

Authors response: Thank you for this suggestion. We agreed that this addition will indeed add value to the manuscript. We added representative plots for the marker CD80/86 (Figure 5, panel A). In addition, we adjusted Figure 3 in order to show the effect of different concentrations of LPS. Both adjustments to the figure will help to better understand how the MFI values in the graphs were derived. 

• Figure legends should consistently state the number of replicates/independent experiments. It is recognised that the nature of work on pigs mean that replication will be limited - but it needs to be explicit.

Authors response: We agree with this notion. The figure legends have been reviewed again and the number of replicates were now added when missing (Figure 3, 4, 7 and S4 Figure).

• There are some minor typos etc:'rpm' instead of 'g' (L153); lack of superscript (L119); Figure S4 is not referred to in the text; the figure panels in S2 are not explained in the legend; what is FCM (L307)?

Authors response: Thank you for noticing these typos. They are now corrected:

- Line 179: ‘rpm’ adjusted to ‘g’

- Line 142: Superscript is now included

- Figure S4: All Figures are referred to now. 

- Figure S2: The figure panels are now explained in the legend.

- Line 192: With FCM I meant Flow Cytometry. As this abbreviation is not commonly used and to avoid confusion it has now been spelled out everywhere in the paper. 

• Clarify what 'remains to be detected' (L399) means? There is no data? Ambiguous data? Conflicting data?

Authors response: To date, there is no data available. Line 399 (now Lines 432 - 434) has been re-written to clarify this. 

• What does 'appeared' (L266) mean?

Authors response: The word ‘appeared’ is not used correctly in this context (too ambiguous). The word appeared has been removed.

• The values of the PCA analysis is unclear. It isn't really explained or described in the Results section (there is some attempt to address this belatedly in Discussion) and it feels a bit of an afterthought.

Authors response: Thank you for the comment. The reason to conduct the principal component analysis (PCA) was to analyse the grouping of these responses of frhBMDCs and cryoBMDCs to different stimulation conditions. We now explain the value of the PCA plot as these values are described and explained more extensively in the results section. The paragraph (Lines 368 - 386) has been re-written. In addition, the legend of Figure 7 has been re-written. To connect the PCA with the previous results, we have now clarified the added value of conducting such an analysis in Lines 368 – 372.

AUTHOR COMMENTS:

We made some minor changes which were detected during the review process. 

• In the article we used ‘Macrogard’ and ‘MacroGard’. The latter brand name is the correct one and we adjusted this throughout the article. 

• In the article we used ‘mean fluorescent intensity’ this should be ‘median fluorescent intensity’.We adjusted this throughout the article.

• The figure legenda’s and the order of the figures are adjusted according to the PLOSONE’s style requirements. 

Looking forward to your decision.

With kind regards and on behalf of the authors,

Mirelle Geervliet

Cell Biology and Immunology Group

Wageningen University

P.O. Box 338

6700 AH Wageningen

The Netherlands

E-mail: geervlietmirelle@gmail.com

Telephone number: +3161657994

References: 

1. Piriou-Guzylack L, Salmon H. Membrane markers of the immune cells in swine: an update. Veterinary research. 2008;39(6):1.

2. Summerfield A, McCullough KC. The porcine dendritic cell family. Developmental & Comparative Immunology. 2009;33(3):299-309.

3. Vreman S, Auray G, Savelkoul HF, Rebel A, Summerfield A, Stockhofe-Zurwieden N. Neonatal porcine blood derived dendritic cell subsets show activation after TLR2 or TLR9 stimulation. Developmental & Comparative Immunology. 2018;84:361-70.

---

## [Decision Letter · Decision Letter 1]

13 May 2020

Differential immunomodulation of porcine bone marrow derived dendritic cells by E. coli Nissle 1917 and β-glucans

PONE-D-19-28012R1

Dear Dr. Geervliet,

We are pleased to inform you that your manuscript has been judged scientifically suitable for publication and will be formally accepted for publication once it complies with all outstanding technical requirements.

With kind regards,

Jagadeesh Bayry, DVM, PhD, HDR

Academic Editor

PLOS ONE

Additional Editor Comments (optional):

Reviewers' comments:

Reviewer's Responses to Questions

**Comments to the Author**

1. If the authors have adequately addressed your comments raised in a previous round of review and you feel that this manuscript is now acceptable for publication, you may indicate that here to bypass the “Comments to the Author” section, enter your conflict of interest statement in the “Confidential to Editor” section, and submit your "Accept" recommendation.

Reviewer #2: All comments have been addressed

Reviewer #3: All comments have been addressed

2. Is the manuscript technically sound, and do the data support the conclusions?

Reviewer #2: Yes

Reviewer #3: Yes

3. Has the statistical analysis been performed appropriately and rigorously? 

Reviewer #2: Yes

Reviewer #3: Yes

4. Have the authors made all data underlying the findings in their manuscript fully available?

Reviewer #2: Yes

Reviewer #3: Yes

5. Is the manuscript presented in an intelligible fashion and written in standard English?

Reviewer #2: Yes

Reviewer #3: Yes

6. Review Comments to the Author

Reviewer #2: Authors have made all suggested changes and improved the manuscript as required. I recommend the editorial board to accept this paper.

Reviewer #3: I made several comments and suggestions on the original submission. I am satisfied that the authors have taken these comments on board and made changes to the manuscript accordingly.

7. PLOS authors have the option to publish the peer review history of their article (what does this mean?). If published, this will include your full peer review and any attached files.

Reviewer #2: No

Reviewer #3: No

---

## [Editor Report · Acceptance letter]

10 Jun 2020

PONE-D-19-28012R1 

Differential immunomodulation of porcine bone marrow derived dendritic cells by E. coli Nissle 1917 and β-glucans 

Dear Dr. Geervliet:

I'm pleased to inform you that your manuscript has been deemed suitable for publication in PLOS ONE. Congratulations! Your manuscript is now with our production department. 

Kind regards, 

on behalf of

Dr. Jagadeesh Bayry 

Academic Editor

PLOS ONE